# Dominant and Priming Role of Waterlogging in Tomato at e[CO_2_] by Multivariate Analysis

**DOI:** 10.3390/ijms232012121

**Published:** 2022-10-11

**Authors:** Rong Zhou, Fangling Jiang, Xiaqing Yu, Lamis Abdelhakim, Xiangnan Li, Eva Rosenqvist, Carl-Otto Ottosen, Zhen Wu

**Affiliations:** 1College of Horticulture, Nanjing Agricultural University, Nanjing 210095, China; 2Department of Food Science, Aarhus University, DK-8200 Aarhus, Denmark; 3Northeast Institute of Geography and Agroecology, Chinese Academy of Sciences, Changchun 130102, China; 4Department of Plant and Environmental Sciences, University of Copenhagen, DK-2630 Taastrup, Denmark

**Keywords:** tomato, waterlogging, elevated CO_2_ concentration, stress memory, multivariate analysis

## Abstract

The frequency of waterlogging episodes has increased due to unpredictable and intense rainfalls. However, less is known about waterlogging memory and its interaction with other climate change events, such as elevated CO_2_ concentration (e[CO_2_]). This study investigated the combined effects of e[CO_2_] and two rounds of waterlogging stress on the growth of cultivated tomato (*Solanum lycopersicum*) and wild tomato (*S. pimpinellifolium*). The aim is to elucidate the interaction between genotypes and environmental factors and thereby to improve crop resilience to climate change. We found that two rounds of treatments appeared to induce different acclimation strategies of the two tomato genotypes. *S. pimpinellifolium* responded more negatively to the first-time waterlogging than *S. lycopersicum*, as indicated by decreased photosynthesis and biomass loss. Nevertheless, the two genotypes respond similarly when waterlogging stress recurred, showing that they could maintain a higher leaf photosynthesis compared to single stress, especially for the wild genotype. This showed that waterlogging priming played a positive role in stress memory in both tomato genotypes. Multivariate analysis showed that waterlogging played a dominant role when combined with [CO_2_] for both the cultivated and wild tomato genotypes. This work will benefit agricultural production strategies by pinpointing the positive effects of e[CO_2_] and waterlogging memory.

## 1. Introduction

Atmospheric CO_2_ concentration (a[CO_2_]) has gradually increased over the last 250 years, and is expected to continue growing in the future with an ever-faster speed [1]. Elevated [CO_2_] (e[CO_2_]) is known to have significant effects on plant growth, as CO_2_ is a primary molecule for carbon assimilation. In general, e[CO_2_] has been found to boost crop growth and improve yield through photosynthesis and other physiological pathways [2].

Extreme rainfall and waterlogging events negatively affected crop production, resulting in enormous economic losses and elevated concerns about food security [3]. Most terrestrial plants including almost all crops are highly vulnerable to waterlogging conditions [3,4]. The frequency and intensity of weather extremes increase, and more and more farming regions are exposed to waterlogging [3]. Therefore, it is urgent to understand how crop plants respond to waterlogging, especially when waterlogging happened more than once.

The e[CO_2_] has diverse ecological and physiological effects on plants that are usually accompanied by other climate change scenarios. The plants growing at e[CO_2_] may be subjected to various abiotic stresses including waterlogging [2]. Previous studies mainly focused on exploring the effects of e[CO_2_] or waterlogging alone on plants [5,6,7,8,9,10]. Only few studies focused on the combined effects of e[CO_2_] and waterlogging on plants [11,12], suggesting the need for further studies. The e[CO_2_] (~760 µmol mol^–1^) increased the photosynthesis, starch content and biomass of tree *Senna reticulata* as compared with a[CO_2_] (~380 µmol mol^–1^) after 90 days, which played a positive role in alleviating the damage to trees caused by waterlogging for 45 days [11]. The decrease in photosynthesis was overcome by the e[CO_2_] (800 µmol mol^–1^) in three rootstocks of sweet cherry, together with increasing soluble sugars and starch and less intensive stress as indicated by lower proline level than a[CO_2_] (400 µmol mol^–1^) for seven days [12]. By comparison, the effects of e[CO_2_] on plants at drought stress and drought ‘memory’ are well-documented [12,13]. Moreover, less studies on the priming and memory effects of waterlogging have been performed [14,15,16], which is an overlooked gap as waterlogging has become one of the most challenging elements of climate change [12].

Tomato (*Solanum lycopersicum*) is an important economic crop, widely grown in the world and has been a model plant for scientific studies. Many research projects focusing on either the effect of e[CO_2_] or waterlogging have been conducted on tomato plants [5,6,8,9]. On one hand, enhanced yield and foliar deformation was seen in tomato grown at high [CO_2_] (1000 ppm) for 16 weeks [6]. On the other hand, waterlogging induced a decrease in the stomatal conductance (g_s_) and transpiration in tomato already after 24 h [5]. The combined physiological, biochemical, and proteomic analyses of tomato plants at waterlogging for short term (24 h, 48 h and 72 h) showed that proteins in both leaf and root played crucial roles in the stress response [8,9]. The growth of tomato plants was significantly and negatively influenced by waterlogging for 14 and 28 days, as indicated by a decrease in seedling height and biomass accumulation [17]. Our previous study found that as compared with a[CO_2_], e[CO_2_] increased photosynthesis and biomass of the cultivated tomato at salinity and combined salinity and waterlogging stress [18]. The combination of e[CO_2_] and waterlogging may lead to altered relations between physiological indexes depending on genotypes. With potentially contrasting effects, it is of both importance and novelty to explore the combined effects of e[CO_2_] and waterlogging priming on tomato.

Photosynthesis is one of the most sensitive processes in plants to abiotic stress including waterlogging [13,19], while leaf pigment and chlorophyll fluorescence is closely related with the activity of leaf photosynthesis. The photosynthesis of the cultivated tomato at salinity and combined salinity and waterlogging stress decreased, while that of the wild tomato at individual salinity, waterlogging and their combination dropped [18]. To improve the understanding of the physiological response of climate change effects on plants, it is critical to explore the responses of photosynthesis and other parameters at leaf-level [19,20]. Hence, herein we analyzed the gas exchange (net photosynthetic rate, P_n_; g_s_; intracellular CO_2_ concentration, C_i_; transpiration rate, E), the related leaf pigment indexes, quenching analysis (non-photochemical quenching, NPQ; fraction of open PSII centers, q_L_; electron transport rate, ETR; quantum efficiency of PSII, F_q_^’^/F_m_^’^), and final biomass of tomatoes. Initially, the plants were treated at a[CO_2_] and e[CO_2_] with or without waterlogging. The plants subjected to the first-round waterlogging treatment were used for the second-round treatment to investigate the effect of waterlogging memory.

We aim to elucidate the interaction between two tomato genotypes and two environmental factors (G × E) and thereby to provide knowledge for the improvement of crop resilience under future climate conditions. Our hypothesis is that (1) the e[CO_2_] can alleviate the damage of waterlogging on tomato; (2) the two genotypes can respond in different way to climatic factor changes due to their diverse genotypes; (3) waterlogging priming could benefit the tomato plants due to stress memory. Improved knowledge on crop physiology towards climate change scenarios will benefit agricultural production strategy.

## 2. Results

### 2.1. The Effects of Waterlogging and e[CO_2_] on Tomato in the First-Round Treatment

During the first-round treatment, the leaf gas exchange, metabolites and plant growth were altered when the plants were exposed to waterlogging and e[CO_2_].

Waterlogging significantly decreased the P_n_ of ‘QX’ at e[CO_2_] from 19.3 ± 1.11 to 13.2 ± 0.17 μmol m^−2^ s^−1^ (Figure 1A). By comparison, the decreased P_n_ of ‘LA’ regardless of [CO_2_] was induced by waterlogging rather than respective controls (Figure 1A). The P_n_ of ‘QX’ at e[CO_2_] was fully recovered at recovery stage to 20.8 ± 0.34 μmol m^−2^ s^−1^, while that of ‘LA’ was not with a low value of 7.7 ± 1.41 μmol m^−2^ s^−1^ (Figure 1A). The trends of g_s_ and E were similar with that of P_n_ in plants at control, waterlogging and recovery (Figure 1B,C). The g_s_ and E of both genotypes during waterlogging were significantly lower than control regardless of [CO_2_], while only that of ‘LA’ did not fully recover (Figure 1B,C). The C_i_ of the two genotypes at waterlogging (251 ± 8.8 and 537 ± 8.2 ppm at a[CO_2_] and e[CO_2_]) were lower than the control (321 ± 0.8 and 695 ± 5.6 ppm at a[CO_2_] and e[CO_2_]) (Figure 1D). Only the C_i_ of ‘QX’ fully recovered to 310 ± 2.5 and 656 ± 17.5 ppm at a[CO_2_] and e[CO_2_] (Figure 1D). Moreover, the e[CO_2_] significantly increased the P_n_ of ‘QX’ and ‘LA’ at control (19.3 ± 1.11 and 17.3 ± 1.14 μmol m^−2^ s^−1^), the P_n_ of ‘QX’ at recovery stage (20.8 ± 0.34 μmol m^−2^ s^−1^) and the P_n_ of ‘LA’ at waterlogging (6.9 ± 0.99 μmol m^−2^ s^−1^) as compared with respective controls at a[CO_2_] (Figure 1A). The e[CO_2_] significantly increased the C_i_ except for the ‘LA’ during recovery (Figure 1D). Four gas exchange parameters were affected by individual genotype and water condition as well as genotype × water, while only P_n_ and C_i_ were affected by [CO_2_] (Table 1). The genotype × [CO_2_] had an effect on P_n_ and C_i_, where a three-way interaction affected P_n_, E and C_i_ (Table 1). The genotype, water, genotype × [CO_2_] and genotype × water showed effects on F_v_/F_m_ (maximum potential quantum efficiency of photosystem II or PSII) (Table 1).

The NBI of ‘QX’ at e[CO_2_] and ‘LA’ at a[CO_2_] at waterlogging were lower than that of ‘QX’ at e[CO_2_] and ‘LA’ at a[CO_2_] at respective controls (Appendix A). Waterlogging decreased the Chl of ‘LA’ regardless of [CO_2_] (Appendix A). The Flav of ‘QX’ during recovery and ‘LA’ at waterlogging was lower than control at e[CO_2_] (Appendix A). The Anth of ‘LA’ increased by waterlogging at a[CO_2_] and at recovery stage regardless of [CO_2_] (Appendix A). The genotype × water affected NBI (Nitrogen balanced index), Chl (chlorophyll index), Flav (flavonol index) and Anth (anthocyanin index) (Table 1).

The plants exposed to waterlogging were smaller in size than controlled plants regardless of [CO_2_] and genotype (Appendix A). Waterlogging significantly decreased the leaf number of ‘LA’ from 7 ± 0.3 to 6 ± 0.0 at a[CO_2_] and from 7 ± 0.3 to 5 ± 0.0 at e[CO_2_] (Figure 2C). Meanwhile, the leaf area and the aboveground FW of both genotypes at waterlogging was smaller than respective controls (Figure 2D,E). For instance, the leaf area of ‘QX’ decreased from 659 ± 64.9 cm^2^ at control to 449 ± 49.6 cm^2^ at waterlogging with a[CO_2_], which decreased from 704 ± 64.1 cm^2^ to 378 ± 23.8 cm^2^ at e[CO_2_] (Figure 2D). Waterlogging significantly decreased the aboveground DW of plants as compared with respective controls, except for ‘QX’ at a[CO_2_] (Figure 2F). Only the genotype had an effect on plant height and internode length, while individual genotype and individual water affected leaf number, leaf area, aboveground FW and aboveground DW (Table 1; Figure 2A,B).

### 2.2. The Effects of Waterlogging Priming and Memory on Tomato in the Second-Round Treatment 

During the second-round treatment, the leaf gas exchange, F_v_/F_m_, chlorophyll fluorescence quenching, metabolites and plant growth of two genotypes under control, waterlogging for the first time and recurring waterlogging were investigated and analyzed.

Generally, the P_n_ of both genotypes at recurring waterlogging was lower than control, but higher than those being suffered for once, regardless of [CO_2_] (Figure 3A). For instance, at e[CO_2_], the P_n_ of ‘QX’ at CC, CW and WW was 20.2 ± 1.39, 7.7 ± 1.33 and 15.1 ± 0.64 μmol m^−2^ s^−1^, respectively. The e[CO_2_] increased the P_n_ of ‘QX’ at control and waterlogging for once and ‘LA’ at control and waterlogging for twice (Figure 3A). At e[CO_2_], the g_s_ and E of both genotypes at recurring waterlogging was lower than control, but higher than CW (Figure 3B,C). The e[CO_2_] decreased the g_s_ and E to different extent in the three treatments (Figure 3B,C). At e[CO_2_], the C_i_ of ‘QX’ at CW was lower than control and recurring waterlogging, while that of ‘LA’ at recurring waterlogging was lower than control and CW (Figure 3D). The e[CO_2_] increased the C_i_ regardless of genotype and [CO_2_] (Figure 3D). The [CO_2_], water, genotype × water, [CO_2_] × water, genotype × [CO_2_] × water had significant effects on the four gas exchange parameters (Table 2).

At a[CO_2_], the F_v_/F_m_ of the first fully expanded leaf in both genotypes at WW was lower than CC (Figure 3E). At e[CO_2_], the F_v_/F_m_ of plants at CW was lower than control, but higher than WW (Figure 3E). The interaction between [CO_2_] and waterlogging affected the F_v_/F_m_ of the first fully expanded leaf (Table 2).

The NPQ of ‘QX’ at CW and WW and ‘LA’ at CW was significantly higher than respective control, regardless of [CO_2_] (Figure 4A). The q_L_, ETR and F_q_^’^/F_m_^’^ in both genotypes at CW and WW significantly decreased as compared with control regardless of [CO_2_], except for the q_L_ of ‘QX’ at a[CO_2_]-WW (Figure 4B–D). The genotype and waterlogging had significant effects on the NPQ, q_L_, ETR and F_q_^’^/F_m_^’^ with the interaction of genotype and water on NPQ, [CO_2_] and water on ETR and F_q_^’^/F_m_^’^ (Table 2).

The NBI of the first fully expanded leaf of both genotypes at CW and WW was lower than control being more pronounced at WW (Table 2, Appendix A). The Chl of the first leaf of ‘QX’ at a[CO_2_]-CW and e[CO_2_]-WW and of ‘LA’ at a[CO_2_]-CW, a[CO_2_]-WW and e[CO_2_]-WW decreased as compared with the controls (Appendix A). The Flav of the first leaf of ‘QX’ at WW was higher than control regardless of [CO_2_], while that from ‘LA’ at e[CO_2_]-WW was higher than control (Appendix A). The Anth of the first leaf of ‘QX’ at WW increased as compared with control and CW, while that from ‘LA’ at CW was higher than control, but lower than WW (Appendix A). The NBI of the last fully expanded leaf from ‘QX’ at CW was lower than control, but higher than WW (Appendix A). The NBI and Chl of the last leaf of ‘LA’ at WW was lower than control, but higher than CW (Appendix A). The Chl of the last leaf of ‘QX’ at CW and WW decreased as compared with control (Appendix A). The Flav of the last leaf of ‘QX’ at WW increased compared with control and CW (Appendix A). The Flav and Anth of the last leaf of ‘LA’ at WW was higher than control, but lower than CW (Appendix A). The genotype and water affected NBI, Chl, Flav and Anth except the Flav for the first and last fully expanded leaf (Table 2).

The plants at CW and WW were smaller with less roots than CC, which was more pronounced at WW (Appendix A). The plant height of ‘QX’ at a[CO_2_]-WW (64.3 ± 4.41 cm) was shorter than a[CO_2_]-CC (77.0 ± 2.34 cm), while that of ‘LA’ at e[CO_2_]-CW (66.1 ± 3.52 cm) and e[CO_2_]-WW (67.2 ± 5.58 cm) was shorter than e[CO_2_]-CC (85.2 ± 8.63 cm) (Figure 5A). The internode length of ‘QX’ at a[CO_2_]-CW (10.0 ± 0.45 cm) was longer than that at e[CO_2_]-CW (8.0 ± 0.15 cm) (Figure 5B). The leaf number of ‘LA’ at CW was more than WW regardless of [CO_2_] (Figure 5C). The leaf area, aboveground FW and aboveground DW of plants at CW was lower than control, but higher than WW regardless of genotype and [CO_2_] (Figure 5D–F). At e[CO_2_], the inflorescence number of ‘QX’ at WW (2 ± 0.0) was significantly lower than CW (3 ± 0.0) (Figure 5G). At a[CO_2_], the inflorescence number of ‘LA’ at WW (3 ± 0.0) was significantly less than control (4 ± 0.6) and CW (4 ± 0.0) (Figure 5G). By comparison, at e[CO_2_], the inflorescence number of ‘LA’ at CW (4 ± 0.6) and WW (4 ± 0.6) was significantly less than control (5 ± 0.6) (Figure 5G).

### 2.3. Hierarchical Clustering and PCA of All Measured Parameters

The PCA was applied to reduce the multi-dimensional dataset to a few components that accounted for the most variation in changes. The changes of physiological parameters and growth indexes in response to two rounds of treatments were explored. In total, 15 parameters from eight treatments and 25 parameters from 12 treatments were included in the PCA for the first- and second-round treatment, respectively (Appendix A). Three principal components explained 89.783% and 83.959% of the variation for the first- and second-round treatment, respectively (Table 3 and Table 4, Appendix A). For the first-round treatment, the first component (PC1), PC2 and PC3 accounted for 53.441%, 22.063% and 14.280% of the observed variation, respectively (Table 3). The P_n_, g_s_, E, NBI, Anth, leaf area, aboveground FW and DW contributed most to the variation in PC1 for the first-round treatment (Table 3). For the second-round treatment, the PC1, PC2 and PC3 accounted for 54.306%, 16.337% and 13.317% of the observed variation, respectively (Table 4). The NBI, Chl, Flav, Anth, q_L_, ETR, F_q_^’^/F_m_^’^, leaf area, aboveground FW and DW contributed most to the variation in PC1 for the second-round treatment (Table 4).

Hierarchical clustering of all measured parameters indicated how the two genotypes were affected differently by the climatic factors at the two rounds of treatments (Figure 6). One dominating cluster appeared defined by the waterlogging treatment, while the control treatment was clustered in the second group when rescaled distance was 10 for the first-round treatment (Figure 6A). In each main group, the treatments with a[CO_2_] and e[CO_2_] except ‘QX’- a[CO_2_]-W was clustered, respectively, as a sub-group when the rescaled distance was 5 (Figure 6A). For the second-round treatment, three dominating clusters were formed when the rescaled distance was 10, wherein those three consisted of all treatments at WW, CW, and CC, respectively (Figure 6B).

## 3. Discussion

### 3.1. Complex and Distinct Response of Two Tomato Genotypes to e[CO_2_] When Interacted with Repeated Waterlog

The e[CO_2_] accelerates plant photosynthesis and improves water-use efficiency unless there are other limiting factors involved [2,6,19]. In accordance, higher P_n_ at e[CO_2_] were observed than at a[CO_2_] at control conditions for both rounds of treatment in tomato. The effects of e[CO_2_] on tomato plants could be altered when other climatic factors become limited; thereby, the interactive effects of multiple environmental factors such as [CO_2_] and water regime need further investigation [21]. For instance, the e[CO_2_] usually improved water-use efficiency of plants at reduced water conditions due to decreased g_s_ [19,22]. Another example was that the positive effects of e[CO_2_] on P_n_ of tomato were observed at moderate drought stress but disappeared at severe drought [13]. In this study, [CO_2_] had less effect on the measured parameters as compared with the other two factors (genotype and water regime) in both rounds of treatment. Meanwhile, [CO_2_] interacted with genotype or water regime or both, which make the responses of plants complicated. The e[CO_2_] increased the P_n_ of ‘QX’ during recovery and ‘LA’ during waterlogging in the first treatment, while the e[CO_2_] boosted the P_n_ of ‘QX’ at CW and ‘LA’ at WW in the second treatment. This corresponds to the fact that e[CO_2_] can ameliorate the negative influence of waterlogging on the Amazonian tree *Senna reticulate* and tomato in terms of P_n_ and biomass [11,18]. The e[CO_2_] have also been shown to overcome the reduction in photosynthesis and growth caused by waterlogging in strawberry rootstocks [12]. However, the increase in P_n_ did not result in more biomass and dry matter accumulation in tomato at e[CO_2_], which was also recently reported in wheat [19]. This finding corresponded in both rounds of treatments. One explanation could be long-term e[CO_2_]-induced photosynthetic acclimation with little or even no improvement of maximum carboxylation velocity of Rubisco (V_cmax_) and maximum electron transport rate (J_max_) at high [CO_2_] [19,23]. Another reason was that the increased source from photosynthesis was not able to transfer to the sink component due to the imbalance between source and sink. Therefore, the e[CO_2_] alleviated the damage on tomato leaf photosynthesis, but not biomass loss caused by waterlogging as a consequence of photosynthetic acclimation and carbohydrate imbalance.

Waterlogging negatively influenced both vegetative and reproductive growth of both the aboveground and underground parts of plants [24], which was verified here in tomato as indicated by both data of measured parameters and plant performance during harvest. Photosynthesis, as one of the main physiological processes, is vulnerable to waterlogging [25,26]. Waterlogging is harmful since it restricted CO_2_ and O_2_ availability and limited gas diffusion in plants [3,10,27]. This explains why the photosynthetic parameters of two tomato genotypes here were severely affected by waterlogging. However, cultivated tomato showed higher photosynthesis under waterlogging in both rounds of treatments than wild tomato. *S. pimpinellifolium* was more sensitive to waterlogging as indicated by more pronounced decrease in the photosynthetic parameters (P_n_, g_s_, E, C_i_) and chlorophyll fluorescence (F_v_/F_m_) than *S. lycopersicum* at waterlogging. The full recovery of photosynthesis in *S. lycopersicum* and partial recovery in *S. pimpinellifolium* also proved the response difference. The adaption of organisms to varying environments happened by two primary mechanisms including genetic variation and phenotypic plasticity [28]. Here, both mechanisms contributed to the different responses, since the two genotypes had an intrinsic genetic background. Meanwhile, the breeding of modern cultivars such as ‘QX’ with more extensive plasticity possibly produced a genotype with high resilience under adverse environments [29].

The reduction in P_n_ in tomato at waterlogging in the first treatment was due to stomatal factor, since the g_s_ and C_i_ simultaneously decreased [30]. Accordingly, waterlogging resulted in low P_n_ with decreased g_s_ and C_i_ in maize [26], which was also caused by stomatal factor. Another reason for the low P_n_ in tomato at waterlogging and recovery stage could be due to the decreased chlorophyll content as shown by Chl index. Ahsan et al., (2007a), have reported the low chlorophyll content in tomato leaves caused by the waterlogging [8]. Manik et al., (2019), concluded that the waterlogging stress reduced photosynthetic activity and g_s_ and decreased chlorophyll synthesis as well as promoted chlorophyll degradation [10]. Moreover, photosystem I (PSI) sensitivity resulted in PSII inactivation through inhibiting electron transport capacity at PSII acceptor [25]. The low level of O_2_ caused by waterlogging can restrict the electron transport as indicated by low ETR in tomato at waterlogging, which could hereby suppress ATP (adenosine-triphosphate) yield and mitochondrial respiration. Overall, the decrease in P_n_ in tomato at waterlogging was the results of stomatal factor, low chlorophyll content and electron transport inhibition.

### 3.2. Waterlogging Priming Enhanced Stress Tolerance and Waterlogging Played Dominant Effects

Both genotypes showed waterlogging memory, since the decrease in photosynthesis was less pronounced in plants being subjected to waterlogging for twice than those for once. The drop of P_n_ at the second treatment of waterlogging was mainly due to the non-stomatal factor as shown by lack of effect in C_i_ [30]. Both genotypes were initially sensitive to waterlogging for the first time as seen by decreased photosynthesis and biomass. By comparison, the plants exposed to waterlogging twice can maintain the plant performance better than those suffered once in terms of photosynthesis, especially for wild genotype. Priming by an eliciting stress can induce memory that enables plants to increase their tolerance to future stress events, which has been suggested to be a promising strategy for plants to cope with the abiotic stresses under climate change scenarios [31]. Pre-treatment or priming approaches to activate the waterlogging tolerance of plants needs further investigation [24]. Our results showed the positive effects of waterlogging priming and stimulating waterlogging memory on tomato physiology. A recent publication on soybean demonstrated that waterlogging priming at a vegetative period for seven days would mitigate the damage of subsequent waterlogging stress at a reproductive period [32]. Likewise, grass species can generate drought memory as shown by improved performance at recurring drought, where upregulation of the antioxidants played crucial roles [16]. Waterlogging-primed wheat had higher activity of enzymes in ascorbate-glutathione cycle and increased expression of proteins related to energy metabolism and stress defense [15] One of reason that *Trichoderma harzianum* alleviates waterlogging-induced growth reduction in tomato was less reduction in anthocyanin [17]. Here, high Anth and flav index in the first fully leaf from both genotypes at recurring waterlogging suggested the positive role of anthocyanin modulating stress priming and memory. Waterlogging priming decreased H_2_O_2_ content, fermentation and alanine aminotransferase activity, but increased the concentration of amino acids, sucrose, and total soluble sugars in both roots and leaves of soybean [32]. Thereby, the increased content of anthocyanin with antioxidant capacity and high levels of antioxidant enzymes resulting in low H_2_O_2_ content could contribute to initiate waterlogging memory and improve tomato tolerance to subsequent stress.

Multivariate analysis and crop model prediction have been used to explore the effects of individual climatic factors, which are challenged when two or more climatic factors are combined due to the non-additive effects [19,33,34]. Meanwhile, research using multifactorial climatic experiments on more than one cultivar are urgently needed [35]. Our study showed that genotype and water regime had significant effects on most parameters at both rounds of treatments. Response strategy difference of tomatoes to the two rounds of treatments was observed in PCA. The photosynthesis and biomass explained most of the variation in PC1 for the first-round treatment, while most of the variation for the second-round treatment was explained by metabolites, quenching parameters, and the biomass. Not only did we find difference between the two rounds of treatments in the indexes responsible for response variation, but also a difference between the climatic factors were noticed. In the first-round treatment, the treatments with waterlogging formed one main cluster, while those with controlled irrigation were grouped, where the two genotypes were distributed in each main clusters. Similarly, in the second-round treatment, there were three dominating groups, where WW, CW and CC were of importance, respectively. The effects of waterlogging were more than that of the genotypes and [CO_2_] on the measured parameters, highlighting a strong genotype-by-environment interaction in tomatoes to waterlogging stress and e[CO_2_].

Genetic differences between the tomatoes’ responses to waterlogging, salinity, e[CO2] and their interaction were observed [18]. Here, we found that the positive effects of e[CO_2_] on tomato photosynthesis, plant growth and biomass accumulation cannot always counteract the negative effects of waterlogging (Figure 7),which is genetic dependent. This is in accordance with our previous study [18], confirming the complexity of the plants’ response to combined stress that cannot be simply speculated from the response to individual stress. By comparison, plants under combined e[CO2] and salinity showed decreased photosynthesis ability with unsignificant change on biomass as compared with a[CO2] [18]. More important, we first found that waterlogging priming and memory can alleviate the damage of waterlogging on photosynthesis as indicated by less decreased P_n_, even though this is based on the biomass penalty (Figure 7). More [CO2] available in the cells of chloroplast and increased evaporation because of high g_s_, enhanced capacity to scavenging reactive oxygen species as shown by high Flav and Anth, enhanced electron transport due to more fraction of open PSII centers together contributed to the positive effects of waterlogging priming on leaf photosynthesis regardless of [CO2] (Figure 7).

## 4. Materials and Methods

### 4.1. Plant Growth Conditions

We included one cultivated tomato ‘QianXi’ (‘QX’, *S. lycopersicum*) from Known-you seed Co. Ltd., China and one wild tomato ‘LA2093′ (‘LA’, *S. pimpinellifolium*) from Tomato Genetics Resource Centre, Davis, California, USA). Seeds of the two genotypes were sown in two cells of a greenhouse at 400 ppm and 800 ppm [CO_2_] (a[CO_2_] and e[CO_2_]), respectively. Average relative humidity (RH) was 65% and average atmosphere temperature was 23 °C in a greenhouse with 15 h/9 h as daytime/nighttime. Light level in the greenhouse depended on natural solar light. Supplementary light was given when the light level was below 150 μmol m^−2^ s^−1^ photosynthetic photon flux density (PPFD). The plants were irrigated with nutrient solution (EC = 2.4 mS/cm, N = 183 ppm, P = 34 ppm, K = 266 ppm, Mg = 41 ppm, Ca = 152 ppm) once per day.

To acquire the seedlings with uniform size, the cuttings of the two genotypes were transferred to pots (9 cm height, 11 cm diameter) from the mother plants. The cuttings were kept at the same [CO_2_] as the stock plants. The 20-days-old tomato cuttings with three to four fully expanded leaves at a[CO_2_] and e[CO_2_] were treated at control (C) and waterlogging (W) condition, respectively, for 5 days as the first-round treatment. There were 14 plants for the treatments of a[CO_2_]-C and e[CO_2_]-C and 7 plants for the treatments of a[CO_2_]-W and e[CO_2_]-W per genotype. Then, the plants recovered (R) for 2 days as a[CO_2_]-R and e[CO_2_]-R. The second-round treatment included (1) CC, control condition for both rounds, (2) CW, control followed by waterlogging, (3) WW, waterlogging for both rounds. Seven plants were used for the treatments of a[CO_2_]-CC, e[CO_2_]-CC, a[CO_2_]-CW, e[CO_2_]-CW, a[CO_2_]-WW and e[CO_2_]-WW per genotype during the second-round treatment. The treatment flow is shown in Figure 8. The control was supplied with normal irrigation while waterlogging was conducted by immersing the pot in water and keeping them in double-layer plastic bags with a water level of 10 cm.

### 4.2. Measurements

The P_n_, g_s_, C_i_, and E of the first fully expanded leaf were measured using a portable photosynthesis system (CIRAS-2, PP Systems, Amesbury, Massachusetts, USA). The measurements were performed on the plants at the first-round treatments for four days, the recovery stage for two days and the second-round treatments for four days in the morning between 8:00 am and 12:00 am. The data was recorded every 10 s until the values of the four above four parameters were stable. The values of the last minute were averaged to obtain the final value for each parameter.

The F_v_/F_m_ was measured using Handy PEA (Hansatech Instrument, King’s Lynn, England) on the plants at the fourth day of the first-round treatments, at the second day of the recovery stage for and at the fourth day of the second-round treatments for. The leaves were dark-adapted for 30 min using leaf clips before the measurements. The NBI, Chl, Flav and Anth were non-destructively measured on plants at the first and second-round treatments after four days using Dualex (FORCE-A, Centre Universitaire Paris Sud, Cedex, France). The first fully expanded leaf at the first-round and second-round treatment were chosen for the measurements.

Plants were dark-adapted for 20 min before the measurements of chlorophyll fluorescence. Non-photochemical quenching (NPQ), fraction of open PSII centers (q_L_), electron transport rate (ETR) and quantum efficiency of PSII (F_q_^’^/F_m_^’^) were measured on the plants at the second-round treatments for four days using PAM-2500 (Walz, Effeltrich, Germany). The first fully expanded leaves were chosen for the measurements.

The plants at the first and second round of the treatment for 5 days were destructively harvested. Plant height and internode length were measured using a ruler and leaf number was counted. Leaves were taken to measure leaf area using leaf area meter (model 3100, LI-COR, Lincoln, NE, USA). Aboveground fresh weight (aboveground FW) and aboveground dry weight (aboveground DW) after drying the samples at 80 °C for two days were detected using a balance. Inflorescence number was counted only during the measurements on the plants at the second round of the treatment.

### 4.3. Data Analysis

A normality test was performed using SPSS 16.0 (SPSS Inc. Chicago, IL) to ensure the data is qualified for analysis of variance (ANOVA). The ANOVA with Tukey’s post hoc test was conducted using SPSS 16.0 with significance accepted at *p* ≤ 0.05. Hierarchical clustering analysis using squared Euclidian distance on treatment × parameter matrix was performed using SPSS 16.0. Principal component analysis (PCA) using principal components was applied using SPSS 16.0. The figures were made based on Excel 2010 (Microsoft Corporation, Richmond, WA, USA).

## Figures and Tables

**Figure 1 ijms-23-12121-f001:**
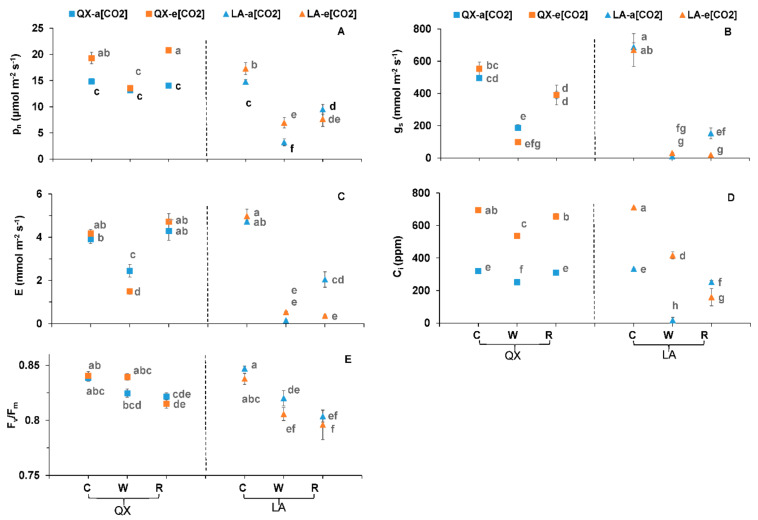
(**A**) Net photosynthetic rate (P_n_), (**B**) stomatal conductance (g_s_), (**C**) transpiration rate (E), (**D**) intracellular CO_2_ concentration (C_i_), and (**E**) maximum quantum efficiency of photo system II (F_v_/F_m_) of the first fully expanded leaf from the two tomato genotypes (‘QX’, *S. lycopersicum* and ‘LA’, *S. pimpinellifolium*) after the first-round treatments. The a[CO_2_] and e[CO_2_] indicated 400 and 800 ppm CO_2_ concentration, respectively. The ‘C’, ‘W’ and ‘R’ followed by a[CO_2_] or e[CO_2_] indicated that the plants were at control and waterlogging condition for four days and recovery stage for two days, respectively. The data indicated average values ± SE (*n* = 3). The ANOVA was performed within all the treatments of two genotypes. Different small letters indicated significant differences (*p* < 0.05).

**Figure 2 ijms-23-12121-f002:**
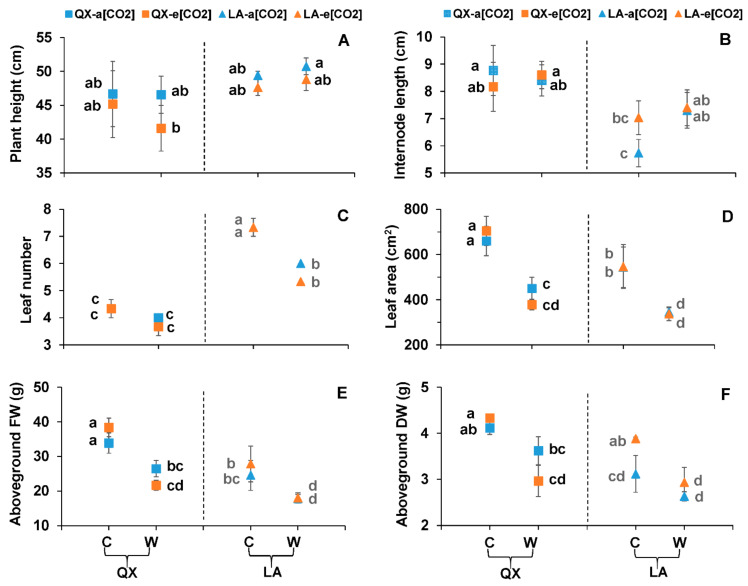
(**A**) Plant height, (**B**) internode length, (**C**) leaf number, (**D**) leaf area, (**E**) aboveground fresh weight (aboveground FW) and (**F**) aboveground dry weight (aboveground DW) of the two tomato genotypes (‘QX’, *S. lycopersicum* and ‘LA’, *S. pimpinellifolium*) after the first-round treatments. The treatments, data analysis and presentation are the same as Figure 1, except the treatments lasted five days.

**Figure 3 ijms-23-12121-f003:**
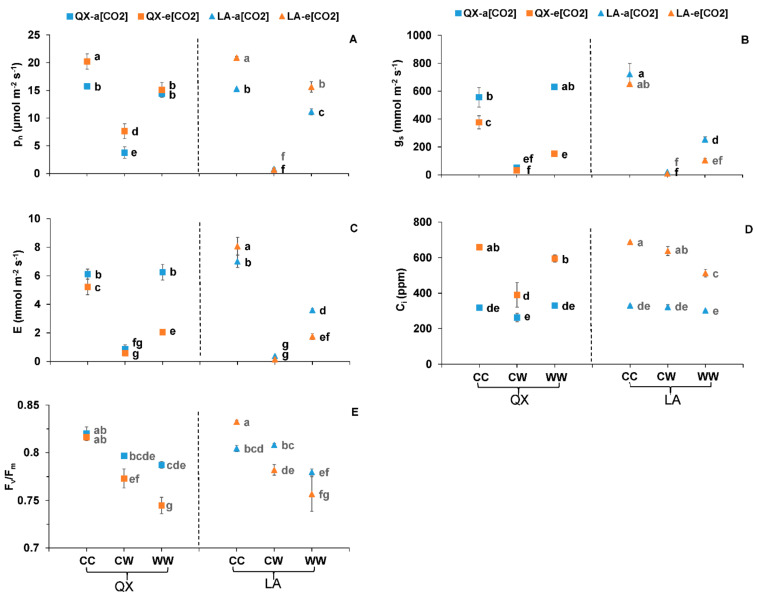
(**A**) Net photosynthetic rate (P_n_), (**B**) stomatal conductance (g_s_), (**C**) transpiration rate (E), (**D**) intracellular CO_2_ concentration (C_i_) and (**E**) maximum quantum efficiency of photo system II (F_v_/F_m_) of the first fully expanded leaf from the two tomato genotypes (‘QX’, *S. lycopersicum* and ‘LA’, *S. pimpinellifolium*) after the second-round treatments for four days. The a[CO_2_] and e[CO_2_] indicated 400 and 800 ppm CO_2_ concentration, respectively. The ‘CC’, ‘CW’ and ‘WW’ followed by a[CO_2_] or e[CO_2_] indicated that the plants were at (1) control condition for both rounds; (2) control followed by waterlogging condition; and (3) waterlogging condition for both rounds, respectively. The data indicated average values ± SE (*n* = 3). The ANOVA was performed within all the treatments of two genotypes. Different small letters indicated significant differences (*p* < 0.05).

**Figure 4 ijms-23-12121-f004:**
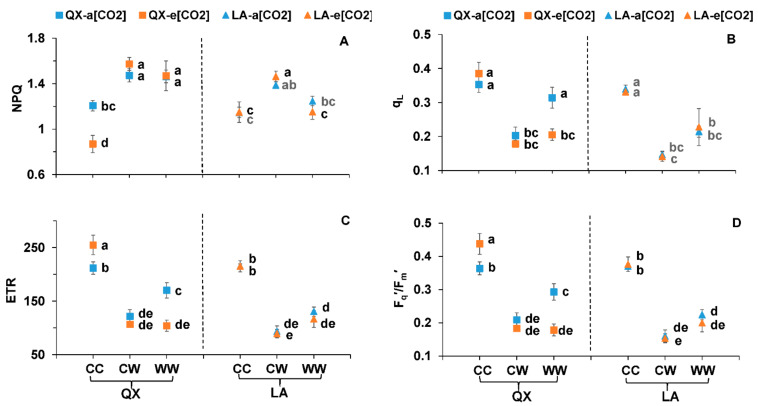
(**A**) Non-photochemical quenching (NPQ), (**B**) fraction of open PSII centers (q_L_), (**C**) electron transport rate (ETR) and (**D**) quantum efficiency of PSII (F_q_^’^/F_m_^’^) of the first fully expanded leaf from the two tomato genotypes (‘QX’, *S. lycopersicum* and ‘LA’, *S. pimpinellifolium*) after the second-round treatments for four days. The treatments, data analysis and presentation are the same as Figure 3.

**Figure 5 ijms-23-12121-f005:**
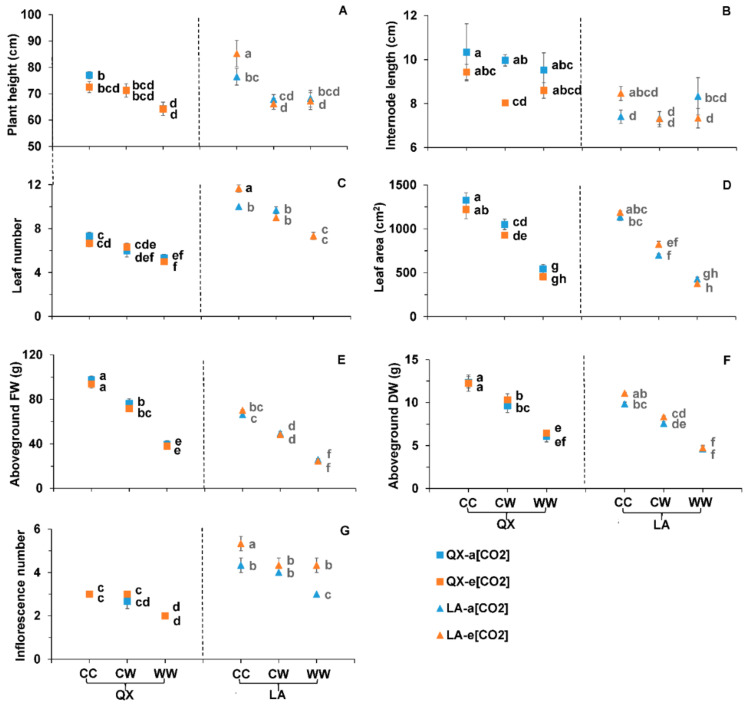
(**A**) Plant height, (**B**) internode length, (**C**) leaf number, (**D**) leaf area, (**E**) aboveground fresh weight (aboveground FW), (**F**) aboveground dry weight (aboveground DW) and (**G**) inflorescence number of the two tomato genotypes (‘QX’, *S. lycopersicum* and ‘LA’, *S. pimpinellifolium*) after the second-round treatments for five days. The treatments, data analysis and presentation are the same as Figure 3.

**Figure 6 ijms-23-12121-f006:**
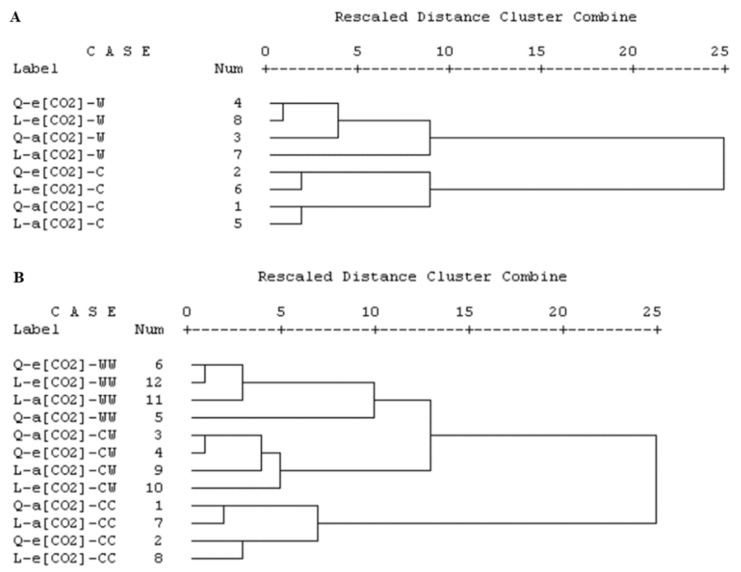
Hierarchical clustering analysis based on Squared Euclidian distance on treatment × parameter matrix of (**A**) the first and (**B**) the second rounds of treatments, respectively. Q and L indicated the two tomato genotypes, ‘QX’, *S. lycopersicum* and ‘LA’, *S. pimpinellifolium*, respectively. The a[CO_2_] and e[CO_2_] indicated 400 and 800 ppm CO_2_ concentration, respectively. In Figure 6A, the C and W indicated that the plants were at control and waterlogging condition for four days, respectively. In Figure 6B, the ‘CC’, ‘CW’ and ‘WW’ followed by a[CO_2_] or e[CO_2_] indicated that the plants were at (1) control condition for both rounds; (2) control followed by waterlogging condition; and (3) waterlogging condition for both rounds, respectively.

**Figure 7 ijms-23-12121-f007:**
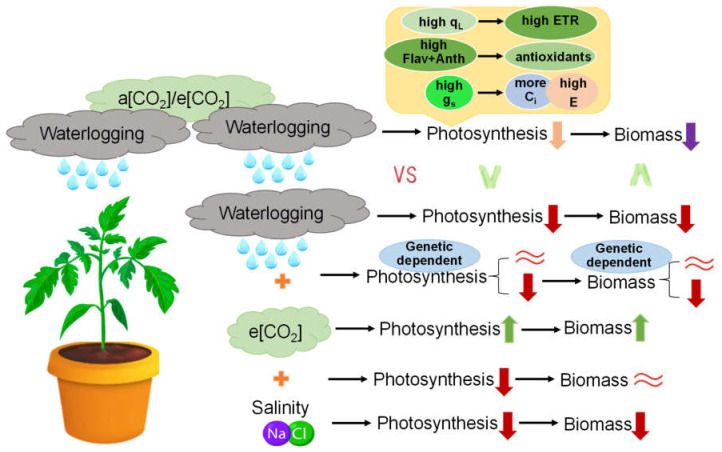
The interaction between waterlogging, salinity and e[CO_2_] on tomato photosynthesis and biomass accumulation. The yellow boxes above the top showed the potential regulatory mechanism on waterlogging memory. Up and down arrows indicated significantly increase and decrease, respectively; 
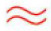
 indicated non-significant change.

**Figure 8 ijms-23-12121-f008:**
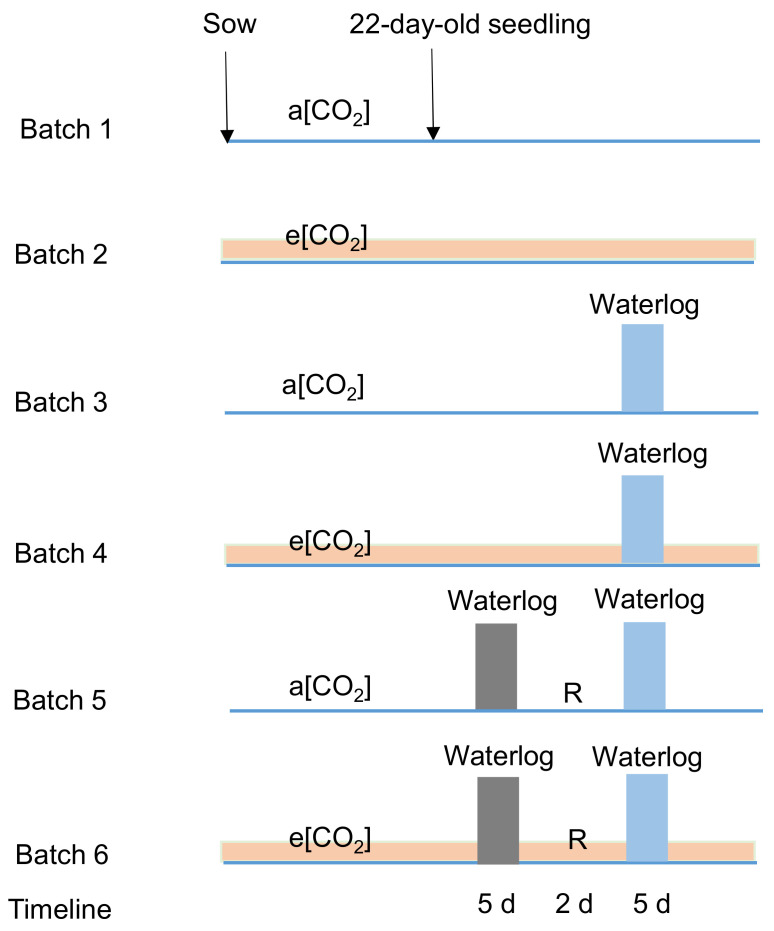
Treatment flow of the current study. The seeds were sown in greenhouse at a[CO_2_] and e[CO_2_]. Then, 22 days after sowing, the first round of treatments started and lasted for five days, followed by two days of recovery stage (‘R’) and the second-round of treatments for five days. The a[CO_2_] and e[CO_2_] indicated 400 and 800 ppm CO_2_ concentration, respectively.

**Table 1 ijms-23-12121-t001:** Three-way analysis of variance (ANOVA) of plant parameters of two tomato genotypes at the first-round treatments with different combinations of genotype (g), CO_2_ concentration ([CO_2_]) and watering condition. The ns, * and ** indicated non-significant change, significant change when *p* ≤ 0.05 and significant change when *p* ≤ 0.01, respectively.

	Main Factors			Interactions			
Parameters	G	[CO_2_]	Water	G × [CO_2_]	G × Water	[CO_2_] × Water	G × [CO_2_] × Water
p_n_	**	**	**	*	**	ns	**
g_S_	**	ns	**	ns	**	ns	ns
E	**	ns	**	ns	**	ns	**
C_i_	**	**	**	**	**	**	**
F_v_/F_m_	**	ns	**	*	*	ns	ns
NBI	**	*	**	ns	**	ns	ns
Chl	ns	*	ns	ns	**	ns	ns
Flav	ns	ns	ns	ns	**	ns	ns
Anth	*	ns	**	ns	**	ns	ns
Plant height	*	ns	ns	ns	ns	ns	ns
Internode length	**	ns	ns	ns	ns	ns	ns
Leaf number	**	ns	**	ns	*	ns	ns
Leaf area	**	ns	**	ns	ns	ns	ns
Aboveground FW	**	ns	**	ns	ns	*	ns
Aboveground DW	**	ns	**	*	ns	*	ns

**Table 2 ijms-23-12121-t002:** Three-way analysis of variance (ANOVA) of plant parameters of two tomato genotypes at the second-round treatments with different combinations of genotype (g), CO_2_ concentration ([CO_2_]) and watering condition. The ns, * and ** indicated non-significant change, significant change when *p* ≤ 0.05 and significant change when *p* ≤ 0.01, respectively.

	Main Factors			Interactions			
Parameters	G	[CO_2_]	Water	G × [CO_2_]	G × Water	[CO_2_] × Water	G × [CO_2_] × Water
p_n_	**	**	**	ns	**	*	**
g_S_	ns	**	**	**	**	**	**
E	ns	**	**	**	**	**	*
C_i_	*	**	**	ns	**	**	**
F_v_/F_m_-first leaf	ns	**	**	ns	ns	**	ns
NPQ	*	ns	**	ns	**	ns	ns
q_L_	*	ns	**	ns	ns	ns	ns
ETR	*	ns	**	ns	ns	**	*
F_q_^′^/F_m_^′^	*	ns	**	ns	ns	**	*
NBI-first leaf	**	ns	**	ns	**	**	ns
Chl-first leaf	**	ns	**	ns	ns	*	ns
Flav-first leaf	ns	ns	**	ns	**	ns	ns
Anth-first leaf	**	*	**	ns	ns	*	ns
NBI-last leaf	**	**	**	ns	**	**	**
Chl-last leaf	**	*	**	ns	**	ns	ns
Flav-last leaf	**	ns	**	ns	**	ns	**
Anth-last leaf	**	*	**	ns	**	ns	ns
Plant height	ns	ns	**	ns	*	ns	ns
Internode length	**	ns	ns	ns	ns	ns	ns
Leaf number	**	ns	**	ns	**	ns	**
Leaf area	**	ns	**	*	ns	ns	ns
Aboveground FW	**	ns	**	ns	**	ns	ns
Aboveground DW	**	ns	**	ns	ns	ns	ns
Inflorescence number	**	**	**	**	ns	ns	ns

**Table 3 ijms-23-12121-t003:** Component matrix of principle component analysis (PCA) result based on the first-round treatments.

	Component		
Parameters	PC1	PC2	PC3
p_n_	0.970	−0.200	−0.079
g_S_	0.905	0.314	0.247
E	0.949	0.194	0.168
C_i_	0.763	−0.109	−0.437
F_v_/F_m_	0.756	−0.045	0.075
NBI	0.861	0.250	0.237
Chl	0.600	0.535	−0.508
Flav	0.460	0.004	−0.808
Anth	−0.842	−0.180	0.418
Plant height	−0.337	0.747	0.490
Internode length	−0.045	−0.962	−0.051
Leaf number	0.145	0.962	0.051
Leaf area	0.849	−0.182	0.490
Aboveground FW	0.808	−0.406	0.387
Aboveground DW	0.816	−0.361	0.271
% of Variance	53.441	22.063	14.280
Cumulative %	53.441	75.503	89.783

**Table 4 ijms-23-12121-t004:** Component matrix of principle component analysis (PCA) result based on the second-round treatments.

	Component				
Parameters	PC1	PC2	PC3	PC4	PC5
P_n_	0.655	0.710	0.096	−0.200	−0.011
g_S_	0.740	0.420	0.177	0.350	0.217
E	0.780	0.453	0.217	0.261	0.187
C_i_	−0.005	0.165	0.470	−0.799	0.221
F_v_/F_m_-first leaf	0.776	−0.327	0.310	0.313	0.145
F_v_/F_m_-last leaf	0.615	0.423	−0.214	0.231	−0.483
NBI-first leaf	0.807	−0.512	−0.204	−0.129	0.000
Chl-first leaf	0.791	−0.260	−0.353	−0.087	0.038
Flav-first leaf	−0.587	0.769	−0.017	0.005	0.078
Anth-first leaf	−0.851	0.486	0.101	−0.063	−0.066
NBI-last leaf	0.954	0.122	−0.132	−0.188	−0.129
Chl-last leaf	0.909	0.307	−0.122	−0.180	−0.124
Flav-last leaf	−0.852	0.029	0.136	0.034	0.434
Anth-last leaf	−0.769	−0.426	0.268	0.185	0.305
NPQ	−0.608	−0.331	−0.427	0.131	−0.065
q_L_	0.889	0.387	0.032	0.062	0.200
ETR	0.928	0.246	0.135	0.039	0.189
F_q_^′^/F_m_^′^	0.929	0.246	0.144	0.038	0.185
Plant height	0.821	−0.154	0.390	0.060	−0.238
Internode length	0.606	0.026	−0.609	0.114	0.264
Leaf number	0.181	−0.246	0.915	0.196	−0.105
Leaf area	0.866	−0.462	0.065	−0.038	0.077
Aboveground FW	0.851	−0.454	−0.173	−0.110	0.074
Aboveground DW	0.846	−0.473	−0.021	−0.186	0.071
Inflorescence number	0.057	−0.186	0.938	−0.036	−0.223
% of Variance	54.306	16.337	13.317	5.313	4.229
Cumulative %	54.306	70.643	83.959	89.272	93.501

## Data Availability

All the data is contained within the article or Appendix A at https://www.mdpi.com/article/10.3390/ijms232012121/s1.

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
