# Peer review of "Dominant and Priming Role of Waterlogging in Tomato at e[CO2] by Multivariate Analysis"

_ijms, 2022, doi:10.3390/ijms232012121_

Round 1

Reviewer 1 Report

Zhou and co-authors have reported the manuscript entitled with “Genotype-Environment Interaction in the Multivariate Analysis of Tomatoes; New Insight in Tomato Resilience to Waterlogging Stress”. The authors investigated combined effects of waterlogging and elevated CO2 on cultivated tomato and wild tomato during the period of plant growth. Although the authors provide interesting data and results which would be utilized for understanding key factors affecting the waterlogging stress and memory, the study should also focus on the molecular base and level of waterlogging responses including ROS and antioxidant enzymes (SOD, POD, CAT) as well as expression levels of ROS, ADH, PDC, and hormones genes with diverse tomato genotypes. Moreover, the manuscript needs to consult with professional and technical editor to be improved for the publication.

Author Response

Responses:

Thanks for the nice comments. We believed that this manuscript provided important information on plant responses under waterlogging conditions. Our study included three factors (tomato genotype, water regime, CO2 concentration) and two rounds of waterlogging were applied to investigate stress memory. Interesting results were found, e.g. complex response of tomato genotypes to e[CO2] in interaction with waterlogging. Moreover, the two rounds of waterlogging treatments appeared to induce different acclimation strategies of the two tomato genotypes.

We totally agree that the molecular responses (e.g. ROS and antioxidant enzymes) and the expression levels of key genes in diverse tomato genotypes under waterlogging will be quite interesting to study. These experiments will be planned in our further studies. However, in this study, we mainly focus on individual and interactive effects of the three factors on photosynthesis, chlorophyll fluorescence quenching and biomass. Due to the large scale of experiment on the factors (3 factors and waterlogging was conducted in two rounds), we already have 6 figures and 4 tables shown in the manuscript. However, we do believe further results concerning ROS and changes of key gene expression will make another good story. We have gone through the whole manuscript and substantial revisions were made especially for the section of results and discussion. Professional and technical editor will definitely join to be sure the manuscript is qualified for the publication.

Reviewer 2 Report

Dear Editors,

After the review of the manuscript entitled “Genotype_Environment Interaction in the Multivariate Analysis of Tomatoes: New insights in Tomato resilience to Waterlogging stress” I have concerns about their publication. The research itself is great with a lot of interesting data but the exposition and representation of them made the read hard and difficult to be follow. For that reason, I recommend their publication after major revision.

The data provided are very interesting since in the current scenario of climate change the understanding and improvement of tolerance to abiotic stress is a significant tool in agriculture. The priming of crops to be more tolerant to a stress such as waterlogging is very interesting, and it should be study more accurately.

That is why, I believe in the importance of reviewing and presenting the data more clearly and making a general conclusion about the results obtained.

Kind regards,

MJose Gonzalo

Author Response

Responses:

We really appreciated your comments. We believed that our manuscript provided interesting and important information on plant responses under waterlogging conditions. Due to the large scale of experiment (3 factors and 2 rounds of treatments) and abundant information, the story is interesting but not easy to tell. Thereby, we have tried to rephase in many spots to better present our results and facilitate the readers. Substantial revisions were made especially for the section of results and discussion. All the revisions have been highlighted in the main text. Please check the revisions if you are interested in. We hope the major revision that we made can satisfy the reviewers.

Reviewer 3 Report

The manuscript by Zhou et al. is a descriptive study comparing the effects of waterlogging and CO2 on two contrasting Solanum genotypes. The study addresses an important question, and the combined analysis of cultivated tomato and its wild relative increases the value of this manuscript.

That said, this manuscript requires advanced proofreading and language editing to increase its legibility. It is difficult to follow and not suitable for publishing in IJMS in its present form. 

Major issues

1) The authors present three hypotheses in the introduction that are more or less based on previous observations (CO2 alleviates the waterlogging damage; priming improves response to waterlogging; two genotypes have contrasting responses). This logical link is lost in the results. I suggest rearranging results and discussions to address these hypotheses.

2) The authors should implement the experiment design and corresponding images into the first part of the Results, introduce the experiment in detail and describe parameters that were monitored to evaluate stress response. As it stands, most of the presented data is just a description of observed changes without meaningful connection that would explain why these parameters are critical. The data presentation would significantly benefit from highlighting the values that are considered optimal and indicating values that represent a severe stress response.  

3) Figures are missing genotype annotation - it is not clear which part represents S. lycopersicum. The comparison of genotypes should be complemented by comparing differences to respective controls. For instance, this would very likely show similarity in all parameters observed in Figure 5. 

4) Statistics. The description of the implemented test is missing. The post-hoc analysis is not described, and data scaling/normalization for PCA is not mentioned. The authors did employ a three-way ANOVA, yet the presented experiment should be evaluated by at least a linear mixed model to elucidate parameters that contributed to the observed differences.   

The presented PCA output is more suitable for supplements. PCAs should be visualized in corresponding figures, not tables. Furthermore, loadings should be complemented by the scores to indicate treatment similarities. 

Minor issues

1) Some portions of the text are overfilled with abbreviations to the extent that the manuscript is nearly illegible:

'The e[CO2] increased the Pn of ‘QX’ at CC and CW condition and ‘LA’ at CC and WW condition (Figure 3A).'

2) Chapter and figure titles should reflect the main message, not the step or methodology. Titles like 'The first-round treatment', 'Hierarchical clustering and PCA of all measured parameters' should be replaced.

Author Response

Reviewer 3

The manuscript by Zhou et al. is a descriptive study comparing the effects of waterlogging and CO2 on two contrasting Solanum genotypes. The study addresses an important question, and the combined analysis of cultivated tomato and its wild relative increases the value of this manuscript. That said, this manuscript requires advanced proofreading and language editing to increase its legibility. It is difficult to follow and not suitable for publishing in IJMS in its present form. 

Responses:

Thanks for the comments. We have tried to rephase in many spots to better present our results and facilitate the readers. All the revisions have been highlighted in the main text. Please check the revisions if you are interested in.

Major issues

1) The authors present three hypotheses in the introduction that are more or less based on previous observations (CO2 alleviates the waterlogging damage; priming improves response to waterlogging; two genotypes have contrasting responses). This logical link is lost in the results. I suggest rearranging results and discussions to address these hypotheses.

Responses:

We would like to give our special thanks to this comment, which help us to improve a lot. So based on this valuable comment, we have re-organized the section especially for discussion. Please check the highlighted parts in the main text, where you will find all the revisions.

In the first sub-title of discussion part, we focused on the alleviated effects of e[CO2] on Pn caused by waterlogging. The paragraph was altered in large scale and ended by ‘Generally, the e[CO2] alleviated the damage on tomato leaf photosynthesis caused by waterlogging depending on genotype and plant age, even though it did not relieve the biomass loss’. Then, the paragraph under the second sub-title focused on the hypothesis on ‘the Distinct response patterns of two tomato genotypes to waterlogging’. Finally, the paragraph under the third sub-title has the central topic on waterlogging priming and memory.

2) The authors should implement the experiment design and corresponding images into the first part of the Results, introduce the experiment in detail and describe parameters that were monitored to evaluate stress response. As it stands, most of the presented data is just a description of observed changes without meaningful connection that would explain why these parameters are critical. The data presentation would significantly benefit from highlighting the values that are considered optimal and indicating values that represent a severe stress response.  

Responses:

Thanks for the comments. Actually, the experimental design and the experiment details for measurements were explained in detail in the section of M and M. This located after the section of discussion, which was based on the criteria of the journal. We do not believe that the description of Methods should be put in the Results.

We have added in the section of introduction as follows: ‘Photosynthesis is one of the most sensitive processes in plants to abiotic stress including waterlogging [13,18], while leaf pigment and chlorophyll fluorescence is closely related with the activity of leaf photosynthesis’. This is to explain the reason why the chosen parameters were detected. Meanwhile, in the section of discussion, we also tried to tell the story by connecting these critical indices. Here are two examples: (1) ‘Taken the different response patterns of NBI, Chl, Flav and Anth index together, the two genotypes showed contrasting responses to waterlogging and its recovery.’ (2) ‘Here, high Anth and flav index in the first fully leaf from both genotypes at recurring waterlogging suggested the positive role of anthocyanin modulating stress priming and memory.’

Since we conducted the ANOVA, based on which, we can scientifically compare the values between all the treatments. What we highlighted is not the value, but the significant difference between the treatments after statistically analyzed the original value. All the values were shown in the figures or tables. But when we described, the significant upregulation or downregulation are paid more attention since it is relatively important than the values themselves. Thereby, we have tried to improve by mentioning ‘normal’ values for control and ‘stressed value’ for waterlogging as follows, according to the above comments. We hope the revision will facilitate the reader. For details, please check the highlighted revision.

Examples:

Waterlogging significantly decreased the Pn of ‘QX’ at e[CO2] from 19.3±1.11 to 13.2 ± 0.17 μmol m-2 s-1 (Figure 1A). By comparison, the decreased Pn of ‘LA’ regardless of [CO2] was induced by waterlogging than respective controls (Figure 1A). The Pn of ‘QX’ at e[CO2] was fully recovered at recovery stage to 20.8±0.34 μmol m-2 s-1, while that of ‘LA’ was not with low value 7.7±1.41 μmol m-2 s-1 (Figure 1A). The trends of gs and E was similar with that of Pn in plants at control, waterlogging and recovery (Figure 1B, 1C). The gs and E of both genotypes during waterlogging were significantly lower than control regardless of [CO2], while only that of ‘LA’ did not fully recover (Figure 1B, 1C). The Ci of the two genotypes at waterlogging (251±8.8 and 537±8.2 ppm at a[CO2] and e[CO2]) were lower than the control (321±0.8 and 695±5.6 ppm at a[CO2] and e[CO2]) (Figure 1D). Only the Ci of ‘QX’ fully recovered to 310±2.5 and 656±17.5 ppm at a[CO2] and e[CO2] (Figure 1D). Moreover, the e[CO2] significantly increased the Pn of ‘QX’ and ‘LA’ at control (19.3±1.11 and 17.3 ±1.14 μmol m-2 s-1), the Pn of ‘QX’ at recovery stage (20.8±0.34 μmol m-2 s-1) and the Pn of ‘LA’ at waterlogging (6.9±0.99 μmol m-2 s-1) as compared with respective controls at a[CO2] (Figure 1A).

The plants exposed to waterlogging was smaller size than controlled plants regardless of [CO2] and genotype (Supplementary Figure 3). Waterlogging significantly decreased the leaf number of ‘LA’ from 7±0.3 to 6±0 at a[CO2] and from 7±0.3 to 5±0 at e[CO2] (Figure 2C). Meanwhile, the leaf area and the aboveground FW of both genotypes at waterlogging was smaller than respective controls (Figure 2D, 2E). For instance, the leaf area of ‘QX’ decreased from 659±64.9 cm2 at control to 449±49.6 cm2 at waterlogging with a[CO2], which decreased from 704±64.1 cm2 to 378±23.8 cm2 at e[CO2] (Figure 2D). Waterlogging significantly decreased the aboveground DW of plants as compared with respective controls, except for ‘QX’ at a[CO2] (Figure 2F).

3) Figures are missing genotype annotation - it is not clear which part represents S. lycopersicum. The comparison of genotypes should be complemented by comparing differences to respective controls. For instance, this would very likely show similarity in all parameters observed in Figure 5. 

Responses:

Thanks for the comments. The explanation was shown in the beginning of M and M. And we have added the information in the caption of figures or tables. Currently the ANOVA was analyzed between all the treatments, and we do compare the genotypes with respective controls, which we have tried to describe in a clearer way.

4) Statistics. The description of the implemented test is missing. The post-hoc analysis is not described, and data scaling/normalization for PCA is not mentioned. The authors did employ a three-way ANOVA, yet the presented experiment should be evaluated by at least a linear mixed model to elucidate parameters that contributed to the observed differences. The presented PCA output is more suitable for supplements. PCAs should be visualized in corresponding figures, not tables. Furthermore, loadings should be complemented by the scores to indicate treatment similarities. 

Responses:

Thanks for the comments. The description of data analysis has been improved based on the comments. Analysis of variance (ANOVA) with Tukey’s post hoc test was conducted using SPSS 16.0 (SPSS Inc. Chicago, IL) with significance accepted at P ≤ 0.05. Hierarchical clustering analysis using squared euclidian distance on treatment × parameter matrix was performed using SPSS 16.0. Principal component analysis (PCA) using principal components was applied using SPSS 16.0. The figures were made based on Excel 2010 (Microsoft Corporation, Richmond, WA, USA). However, the data for PCA was just transferred to SPSS without specific normalization. And we agreed that a linear mixed model could contribute to the observed differences. As we can see, we already have so much information to show. On the other hand, it is fine to focus on the results from ANOVA instead of applying model (Zhou et al., 2020; 2022). And we would like to keep the tables for the PCA as we did in our previous papers (Eller et al., 2020; Zhou et al., 2022).

Reference

Eller, F., Hyldgaard, B., Driever, S. M., & Ottosen, C. O. (2020). Inherent trait differences explain wheat cultivar responses to climate factor interactions: New insights for more robust crop modelling. Global Change Biology, 26(10), 5965-5978.

Zhou, R., Hongjian Wan, Fangling Jiang, Xiangnan Li, Xiaqing Yu, Eva Rosenqvist, Carl-Otto Ottosen (2020). The alleviation of photosynthetic damage in tomato under drought and cold stress by high CO2 and melatonin. International Journal of Molecular Sciences, 21: 5587.

Zhou, R., Yu, X., Song, X., Rosenqvist, E., Wan, H., & Ottosen, C. O. (2022). Salinity, waterlogging, and elevated [CO2] interact to induce complex responses in cultivated and wild tomato. Journal of Experimental Botany.

Minor issues

1) Some portions of the text are overfilled with abbreviations to the extent that the manuscript is nearly illegible:

'The e[CO2] increased the Pn of ‘QX’ at CC and CW condition and ‘LA’ at CC and WW condition (Figure 3A).'

2) Chapter and figure titles should reflect the main message, not the step or methodology. Titles like 'The first-round treatment', 'Hierarchical clustering and PCA of all measured parameters' should be replaced.

Responses:

  • We have tried to improve by applying less abbreviations to help the reader to understand better and faster.
  • Chapter and figure titles has been improved.

Please check the detailed revision in the main text, which were highlighted.

Round 2

Reviewer 1 Report

I would like to thank the authors for improving the manuscript. However, the manuscript still needs to be revised for the clarity of grammar and structure with proofreading and editing service. In particular, the study can focus on more molecular bases including chlorophyll biosynthesis, photosynthesis, and ROS in the tomato genotypes under waterlogging and elevated CO2 concentration.

Author Response

Response:

Thanks for the comments. Firstly, the language of the MS was carefully checked and improved. Secondly, the focus of the current work was physiological response, which we have talked with topic editor that this MS fit the special issue, even though it has no molecular section. However, in our future experiment, more molecular based will be added, even though we can not change the current paper since the work has been conducted. For all the changes, please check the MS.

Reviewer 2 Report

Dear Authors,

I appreciate the work done by the authors to improve the manuscript. Unfortunately, except that from some points are a bit clearer, the problem with the manuscript remains the results communication and the conclusion of the manuscript.

There is a huge data amount but the analysis is not really conclusive.

I think that to be published the manuscript needs a very strong transformation in data interpretation.

The main problems of the manuscript are listed below:

1.- In the Introduction:

            L67-82: I recommend the organizations of paragraphs.

·         The use of wild species as source of tolerances is very well documented in tomato and it is a good and widespread breeding practice.. Is there any evidence of the better /worse response of wild tomato to waterlogging or CO2??

·         L71-79: This paragraph is not related to the wild species use. I would start a new paragraph for this information

2.- Results:

·         The denomination of experiments as first round, second round etc….is not appropriate to understand the research. I would change to another name easier to follow.

·         Graphics are confusing. The representation of the measurements in the first round and second round is different. In my opinion, each measurement should be represented with the same type of graphs to see if there are any differences.

I see a very similar pattern for the treatments and not much difference from the control. I not sure if it is because of the graphics effect or it is like that. That is why it is important to homogenize the graphics by measurent to be able to compare.

The main differences are found between the genotypes, but only a few measurements show a better response than the control. The results must be organized so that they are displayed in an easy and quick way to lighten the text.

·         The PCA analysis should be graphic represented not in graphic.

3.- Discussion:

·         Lines 292-294: “Generally, the e[CO2] alleviated the damage on tomato leaf photosynthesis caused by waterlogging depending on genotype and plant age, even though it did not relieve the biomass loss”. It is not a strong conclusion. Small contribution to the study

·         Lines 324-325: Similar to the other paragraph, the conclusion should add something not just say they are different.

·         Lines 365-367:  Similar to the other paragraph, the conclusion should add something not just say they are different.

The conclusion of each comparison is not strong. In my opinion, there is a lack of final conclusion, because it seems to me that it ends without a clear and direct take home message.

Author Response

Response:

Thanks for the comments. The paragraph was re-organized. The section on explanation of wild tomato usage was removed, and the new paragraph describing the experimental design was provided. We have tried to add figure 7 in the section of M and M, so that the reader can be clear about the experimental design. We added the principal component analysis figures in the supplementary figure 6 to facilitate the reader. The whole MS was improved by carefully checking.

Reviewer 3 Report

I appreciate that the authors did try to update the manuscript, but I don't believe that any of my suggestions were taken seriously, including professional proofreading and editing. The revision did not significantly improve the grammar, legibility, or structure.  

I find the absence of data integration and visualization critical, and the missing statistical evaluation is hardly compensated by pointing out that a similar study has been successfully published in EEB. Noting that, it is surprising that the study with such a similar topic (Salinity, waterlogging, and elevated [CO2] interact to induce complex responses in cultivated and wild tomato), identical structure, and content is not referenced in the submitted manuscript, and any data comparison that would explain novelty or indicate differences in conclusions is missing. I am concerned that the evaluation of S. pimpinelliofolium and its response to waterlogging under elevated CO2 has been published in the above-mentioned manuscript. 

Additional comments to responses to review:

The comparison of genotypes should be complemented by comparing differences to respective controls. For instance, this would very likely show similarity in all parameters observed in Figure 5. 

Currently the ANOVA was analyzed between all the treatments, and we do compare the genotypes with respective controls, which we have tried to describe in a clearer way.

Relative differences in growth (and similar parameters) are as important as the absolute values. The applied data visualization and statistics highlight differences in the stress response profile that would likely be identical once normalized to the corresponding controls.

However, the data for PCA was just transferred to SPSS without specific normalization. 

That is an incorrect approach that is limiting PCA output to the parameters with the highest values, and the contribution of 'small' numbers is absent.

Author Response

Reviewer 3

I appreciate that the authors did try to update the manuscript, but I don't believe that any of my suggestions were taken seriously, including professional proofreading and editing. The revision did not significantly improve the grammar, legibility, or structure. 

I find the absence of data integration and visualization critical, and the missing statistical evaluation is hardly compensated by pointing out that a similar study has been successfully published in EEB. Noting that, it is surprising that the study with such a similar topic (Salinity, waterlogging, and elevated [CO2] interact to induce complex responses in cultivated and wild tomato), identical structure, and content is not referenced in the submitted manuscript, and any data comparison that would explain novelty or indicate differences in conclusions is missing. I am concerned that the evaluation of S. pimpinelliofolium and its response to waterlogging under elevated CO2 has been published in the above-mentioned manuscript.

Response:

The two experiments were conducted separately and written at the same time even though the one in JXB was accepted early. The focus of our previous paper in JXB was the interaction between salt, waterlogging and CO2, while in the study we have new focus on waterlogging memory. We have added this reference in the list. The recent publication on waterlogging memory in soybean was added as well.

Additional comments to responses to review:

The comparison of genotypes should be complemented by comparing differences to respective controls. For instance, this would very likely show similarity in all parameters observed in Figure 5.

Currently the ANOVA was analyzed between all the treatments, and we do compare the genotypes with respective controls, which we have tried to describe in a clearer way.

Relative differences in growth (and similar parameters) are as important as the absolute values. The applied data visualization and statistics highlight differences in the stress response profile that would likely be identical once normalized to the corresponding controls.

However, the data for PCA was just transferred to SPSS without specific normalization.

That is an incorrect approach that is limiting PCA output to the parameters with the highest values, and the contribution of 'small' numbers is absent.

Response:

The reviewer’s comments confused us. In figure 5, the data was compared between all the treatments as well. The comparison between one genotype or both genotypes have both advantage and disadvantages. By comparing between both genotypes at all the treatments, we can not only obtain the changes of one genotype at all the treatments (we can still compare the plants grown under control and stress, which is what we do in most of cases), but also check the difference between genotypes. If we only compared the treatments within one genotype as suggested by reviewer, the statistical analysis between the two genotypes will be missing. Then, how we can check the difference between two genotypes become another problem.

The relative differences are interesting to add, but we believe the original results (like plant height, plant fresh weight, leaf area) are valuable and intuitive to follow. The most important thing is whether there is significant difference between the treatments, instead of how big increase or decrease it is. It is not statically changed between the treatments if there is no significant difference based on ANOVA, even if there seems to have big increase or decrease.

The results of PCA output are correct since we applied the analysis using SPSS, where we checked that the original variables were automatically standardized. We are sorry for the misunderstanding. Thereby, the subsequent PCA analysis output results were based on the standardized variables and there is no need to perform the standardization operation separately here. And if we checked the results, the contribution of ‘small’ numbers is not absent.

Round 3

Reviewer 1 Report

I would like to thank the authors for significantly improving the language of the manuscript.

Author Response

Thanks a lot for your valuable comments, which really help us a lot to improve the quality of our manuscript (MS). The MS has been carefully checked based on all the comments from reviewers. 

Reviewer 2 Report

I would like to thank the authors for the effort to improve the manuscript.

Some of the questions have been really improved. However, the graphical represntation is still a little confusing.

If I am not mistaken, Supplementary Figs 1and 3 are the graphical representation of the individual analysis of each genotype compared to its own controls. The information provided by this comparison is relevant and should be made clear in the figure titles and in the main text paragraphs (line 119 for Supplementary Figure 2). A sentence at the beginign of each paragrpah specifiing the analyis would facilitate the interpretation of the results and differentiate the individual analysis from the joint one.

Moreover, the order of the data in the main text is also important. I would recommend to start with the individual analysis of each genotype against its control and then the joint analysis to see the genotype x treatment interaction.

Author Response

Response to reviewer 2

Comments:

I would like to thank the authors for the effort to improve the manuscript.

Some of the questions have been really improved. However, the graphical represntation is still a little confusing.

If I am not mistaken, Supplementary Figs 1 and 3 are the graphical representation of the individual analysis of each genotype compared to its own controls. The information provided by this comparison is relevant and should be made clear in the figure titles and in the main text paragraphs (line 119 for Supplementary Figure 2). A sentence at the beginning of each paragrpah specifiing the analyis would facilitate the interpretation of the results and differentiate the individual analysis from the joint one.

Moreover, the order of the data in the main text is also important. I would recommend to start with the individual analysis of each genotype against its control and then the joint analysis to see the genotype x treatment interaction

Responses:

We really appreciate your valuable comments. We have tried our best to improve the quality of our manuscript. The following paragraphs are our point-to-point responses.

As we mentioned in Figs 1 and 3, ANOVA was performed within all the treatments. The data analysis of Supplementary Figs 1 and 3 were the same as Figs 1 and 3, respectively. Thereby, actually, the comparison was made between all the treatments of two genotypes. We have tried to make this point clear in the caption of Figs 1 and 3. ‘All the ANOVA was performed within all the treatments of two cultivars.’ Was added Figs 1 and 3 and Supplementary Figs 2 and 4. Line 119 has been rephrased to make it clear how the comparisons were made.

We have added a sentence as a start of some paragraphs to explain the analysis and facilitate the reader to follow. For instance, ‘During the first-round treatment, the leaf gas exchange, metabolites and plant growth were altered when the plants were exposed to waterlogging and e[CO2].’ Was added under the first subtitle of results section. ‘During the second-round treatment, the leaf gas exchange, Fv/Fm, chlorophyll flu-orescence quenching, metabolites and plant growth of two genotypes under control, waterlogging for the first time and recurring waterlogging were investigated and ana-lyzed.’ Were added under the second subtitle of results section.

And to better clarify the effects of genotype × environment (G × E), we performed the ANOVA within all the treatments of the two genotypes. In this way, we can 1) check the effects of individual factor and whether the factors interacted on the parameters; 2) compare the difference between all the treatments within one genotype, e.g. control and waterlogging; 3) compare the difference between two genotypes at the same treatment, e.g. ‘QX’ and ‘LA2093’ at control. If we only perform ANOVA within the treatments of one genotype, it will not be possible for us to compare between the two genotypes. But I would say, there is no right or wrong way, but the suitable way depending on different people’s choice.

Based on your valuable comments, we have re-organize the data in the main text to facilitate the reader. And we really appreciate your valuable comments again.

Reviewer 3 Report

Response:

The two experiments were conducted separately and written at the same time even though the one in JXB was accepted early.

- The manuscript in JXB was accepted in February, at least four months prior this manuscript submission to IJMS

The reviewer’s comments confused us....

- The approach per se is not incorrect, but given the size of the experiment and the very low number of replicates, some interpretations are questionable. The comparison of normalized values would highlight similar or contrasting responses to stressors and support that with statistics. However, I understand that the authors do not want to do that, given the huge differences in reported SE that could undermine the results. In this aspect, I would like to point out that it seems that the normality test is missing (Materials & Methods).    

Comments to the revision:

The legibility of the manuscript has slightly improved, and some comments were addressed. 

The manuscript is still saturated with plots that are difficult to follow, and the structure of the manuscript has not been revised. The manuscript could serve as a data resource for future analyses, but the authors did a poor job in presenting their work. 

The description of employed statistics is mediocre and not reproducible without the specified software tools. That should not be the case for standard analyses, but, unfortunately, it is comparable to other published materials.  

Requested modifications:

I don't believe that there is any chance that the authors would be persuaded to modify figures or the manuscript structure. Thus, I ask them to provide one more paragraph summarizing the results of this manuscript and comparing those with their published study in JXB. Please, supplement this final chapter with a graphical visualization to make that more appealing and to improve the impression of the whole work.  

Author Response

Response to reviewer 3

Comments:

- The approach per se is not incorrect, but given the size of the experiment and the very low number of replicates, some interpretations are questionable. The comparison of normalized values would highlight similar or contrasting responses to stressors and support that with statistics. However, I understand that the authors do not want to do that, given the huge differences in reported SE that could undermine the results. In this aspect, I would like to point out that it seems that the normality test is missing (Materials & Methods).    

Comments to the revision:

The legibility of the manuscript has slightly improved, and some comments were addressed. 

The manuscript is still saturated with plots that are difficult to follow, and the structure of the manuscript has not been revised. The manuscript could serve as a data resource for future analyses, but the authors did a poor job in presenting their work. 

The description of employed statistics is mediocre and not reproducible without the specified software tools. That should not be the case for standard analyses, but, unfortunately, it is comparable to other published materials.  

Requested modifications:

I don't believe that there is any chance that the authors would be persuaded to modify figures or the manuscript structure. Thus, I ask them to provide one more paragraph summarizing the results of this manuscript and comparing those with their published study in JXB. Please, supplement this final chapter with a graphical visualization to make that more appealing and to improve the impression of the whole work.  

Responses:

We really appreciate your valuable comments.

We agreed with the reviewer that the comparison of normalized values may highlight the response. And as the reviewer mentioned that the approaches currently were not incorrect. So based on the reviewers’ comments, we have tried to improve. The normality test is added and mentioned in the M and M to improve, which was shown partly in the response letter in the format of word.

We have tried to improve the figures by adding a dash line between two genotypes and the explanation of x axis was remarked with different genotype. We believe this can facilitate the reader to follow to some extent. We have provided one last graph and one more paragraph to summarize the results and conclusion of the current research together with our results from JXB. Thanks again for all the valuable comments to help us to improve the current MS. For all the detailed revision, please check updated files. We really hope the current MS is qualified and satisfied you.
